# A Protein–Protein Interaction Analysis Suggests a Wide Range of New Functions for the p21-Activated Kinase (PAK) Ste20

**DOI:** 10.3390/ijms242115916

**Published:** 2023-11-02

**Authors:** Ifeoluwapo Matthew Joshua, Meng Lin, Ariestia Mardjuki, Alessandra Mazzola, Thomas Höfken

**Affiliations:** 1Division of Biosciences, Brunel University London, Uxbridge UB8 3PH, UK; ifeoluwapo.joshua@brunel.ac.uk (I.M.J.);; 2Institute of Biochemistry, Kiel University, 24118 Kiel, Germany; 3Department of Biopathology and Medical and Forensic Biotechnologies, University of Palermo, 90133 Palermo, Italy

**Keywords:** p21-activated kinase (PAK), Ste20, budding yeast, *Saccharomyces cerevisiae*, split-ubiquitin, protein–protein interaction, glucose metabolism, pseudohyphal growth, gene expression

## Abstract

The p21-activated kinases (PAKs) are important signaling proteins. They contribute to a surprisingly wide range of cellular processes and play critical roles in a number of human diseases including cancer, neurological disorders and cardiac diseases. To get a better understanding of PAK functions, mechanisms and integration of various cellular activities, we screened for proteins that bind to the budding yeast PAK Ste20 as an example, using the split-ubiquitin technique. We identified 56 proteins, most of them not described previously as Ste20 interactors. The proteins fall into a small number of functional categories such as vesicle transport and translation. We analyzed the roles of Ste20 in glucose metabolism and gene expression further. Ste20 has a well-established role in the adaptation to changing environmental conditions through the stimulation of mitogen-activated protein kinase (MAPK) pathways which eventually leads to transcription factor activation. This includes filamentous growth, an adaptation to nutrient depletion. Here we show that Ste20 also induces filamentous growth through interaction with nuclear proteins such as Sac3, Ctk1 and Hmt1, key regulators of gene expression. Combining our observations and the data published by others, we suggest that Ste20 has several new and unexpected functions.

## 1. Introduction

p21-activated kinases (PAKs) are important signaling molecules that are highly conserved from yeast to man [1,2,3]. These serine/threonine kinases are activated by the Rho GTPases Cdc42 and Rac. The budding yeast *Saccharomyces cerevisiae* expresses three PAKs, Ste20, Cla4 and Skm1, all of them downstream effectors of Cdc42 [4]. Very little is known about Skm1. In contrast, numerous functions have been attributed to Cla4, ranging from septin organization to vacuolar inheritance [4].

The third member of the budding yeast PAK family, Ste20, was initially identified as an activator of three distinct mitogen-activated protein kinase (MAPK) cascades, mediating pheromone response [5,6], filamentous growth [7,8] and the hyperosmotic stress response [9,10]. In all three pathways, Ste20 phosphorylates and thereby activates Ste11, the most upstream component of these MAPK modules [11,12]. Triggering the MAPK cascades eventually results in a change of gene expression.

In the last two decades, other Ste20 functions were identified including the regulation of apoptosis [13] and the modulation of V-ATPase activity [14]. Over time, it emerged that Ste20 is involved in quite diverse and sometimes seemingly unrelated processes. Notably, this has also been observed for PAKs in other eukaryotes [1,2,3]. The wide range of Ste20 functions is also reflected in its subcellular localization. Ste20 can be found in the cytoplasm and at the plasma membrane at sites of polarized growth at the tips of buds and mating projections [15,16]. Furthermore, Ste20 can translocate to the nucleus [13,17], and it also associates with vacuolar membranes [14].

Our knowledge of Ste20 biology has increased tremendously with the identification of some downstream effectors and elucidation of molecular mechanisms. Nevertheless, large gaps in our understanding of Ste20 processes remain. It is, for example, still not entirely clear how Ste20 integrates its diverse cellular activities. Furthermore, for some processes, such as mitotic exit, a crucial role for Ste20 has been demonstrated but the underlying molecular mechanisms remain elusive [18,19,20]. The identification and characterization of more proteins that physically interact with Ste20 would improve our understanding of the functional complexity of Ste20.

Here, we report the results of a genome-wide split-ubiquitin screen for the isolation of proteins that bind to Ste20. In total, 56 proteins were identified, including only a few known Ste20 interactors. Interactions between Ste20 and glucose metabolism enzymes and nuclear proteins were further analyzed. The results suggest a wide range of unexpected functions for Ste20, in particular in many aspects of gene expression.

Due to the high conservation of PAKs, our results may also be relevant for PAKs in higher eukaryotes. In humans, PAKs are implicated in cancer, infectious diseases, diabetes, neurological disorders and cardiac diseases [21,22]. Because of these links, a better understanding of PAK biology becomes increasingly important.

## 2. Results

### 2.1. Overview of the Ste20 Split-Ubiquitin Screen

In order to improve our understanding of Ste20 functions and mechanisms, we performed a split-ubiquitin screen to identify proteins that bind to Ste20. The split-ubiquitin technique is based on the ability of the N-terminal and C-terminal domains of ubiquitin to form a quasi-native ubiquitin (Figure 1) [23,24]. If two proteins, which are attached to the N-terminal and C-terminal portion, respectively, bind to each other, the ubiquitin halves may be forced into proximity and a ubiquitin-like molecule assembles. Ubiquitin-specific proteases, present in the cytoplasm and nucleus, recognize the reconstituted ubiquitin but not its individual halves, and cleave off a reporter that is attached to the C-terminal ubiquitin portion. The technique described here employs a modified version of Ura3, an enzyme essential for uracil biosynthesis, as reporter [24]. This Ura3 variant carries an additional arginine at the extreme N-terminus (RUra3). RUra3 attached to the C-terminal half of ubiquitin is stable and functional. In contrast, RUra3 cleaved off by a ubiquitin-specific protease is rapidly degraded because arginine is a destabilizing residue in the N-end rule pathway. Therefore, the interaction between two proteins fused to the N-terminal and C-terminal halves of ubiquitin results in uracil auxotrophy. Conversely, growth on 5-fluoroorotic acid (5-FOA) indicates a protein–protein interaction because Ura3 converts 5-FOA into the toxic compound 5-fluorouracil.

The split-ubiquitin technique has a number of advantages that make it the ideal tool for the identification of Ste20 interactors. This method is suitable for monitoring interactions with membrane proteins [23,24,25] which is critical because Ste20 associates with the plasma membrane and vacuolar membranes [14,15,16]. The split-ubiquitin assay can also detect weak and transient interactions in vivo [23,25], which allows for the isolation of substrates of the kinase Ste20.

In the screen, we identified 56 proteins as putative interactors of Ste20 (Table 1). Five proteins have been shown by other groups to bind to Ste20 (Table 2). Out of these, the interaction between Bem1 and Ste20 is well established (Table 2). Like Ste20, Bem1 has a role in the three MAPK pathways’ regulating pheromone response [26,27], filamentous growth [28] and hyperosmotic stress response [28]. As a central scaffold protein, Bem1 not only binds to Ste20 but also to the related PAK Cla4 and their upstream activator Cdc42 [26,29,30,31]. Large-scale screens have also revealed that Mlc1 and Ssb2 physically interact with Ste20, and that Spb1 and Ubx7 are phosphorylated by Ste20 (Table 2). Notably, the Bem1-Ste20 and Mlc1-Ste20 interactions observed by others were also detected using the split-ubiquitin technique (Table 2). The fact that other groups independently isolated some of the proteins described here indicates that this approach robustly identifies Ste20 interactors.

We have previously characterized the interactions between Ste20 and five proteins identified in this screen (Sut1, Ncp1, Cbr1, Erg4, Vma13) (Table 2). Importantly, it has also been shown that Ste20 modulates the processes these proteins mediate, namely sterol biosynthesis (Erg4, Cbr1 and Ncp1) [32,33], sterol uptake (Sut1) [17] and V-ATPase activity (Vma13) [14]. Taken together, we present here 46 putative Ste20 interactors not described previously.

To find out whether these interactions are physiologically relevant, we further analyzed them by including published observations. First, the interactions identified in the screen presented here seem to be highly specific. We performed another split-ubiquitin screen using the same library and Rdi1 as bait (Appendix A). We have chosen Rdi1 because, like Ste20, Rdi1 binds to Cdc42 [34,35]. However, Rdi1 is not a downstream effector of Cdc42. Instead, Rdi1 regulates the localization of Cdc42 by extracting it from membranes [34,36]. Rdi1, therefore, like Ste20, localizes to the cytoplasm, the plasma membrane and vacuolar membranes [34,36,37]. Fifty different proteins have been identified in the Rdi1 split-ubiquitin screen (Appendix A). Only one protein (Mlc1) came up in both screens, even though Ste20 and Rdi1 both function around Cdc42 and both localize to the same compartments. This demonstrates that the interactions described here are highly specific. Using the split-ubiquitin assay, Ste20 and Rdi1 physically interact with a very different set of proteins.

The localization of the proteins identified here also suggests that the interactions are real. Importantly, all of the Ste20 interactors can be found in compartments where they could physically interact with Ste20. The huge majority of proteins localize to the cytoplasm and/or the nucleus (Table 1). Some proteins that associate with the plasma membrane, and the membranes of the endoplasmic reticulum (ER) and vacuoles, are peripheral membrane proteins on the cytoplasmic side, such as the aforementioned Bem1 and the vacuolar V-ATPase subunit Vma13 [26,38]. For other proteins, at least a portion of the protein faces the cytoplasm. For example, a topology analysis of Ale1, an enzyme involved in glycerophospholipid biosynthesis, has shown that the protein spans the ER membrane multiple times [39]. However, Ale1 also contains several cytoplasmic loops and a cytoplasmic C-terminus that is over 100 residues long. These proteins would, therefore, be accessible for Ste20.

For three proteins (Ima5, Kin82 and Pps1), subcellular localizations are not known. However, considering their functions and physical interactions with other proteins, it seems very likely that these proteins localize to the cytoplasm and/or nucleus. For example, Pps1 is involved in DNA replication [40] and physically interacts with seven proteins (Ade13, Ccr4, Dhh1, Isw1, Mpt5, Nab2 and Ser3) all of which localize to the nucleus and/or cytoplasm [41,42,43,44,45,46]. It is, therefore, reasonable to assume that Pps1 can also be found in the nucleus and/or cytoplasm.

Ste20 plays key roles in filamentation, the pheromone response and the hyperosmotic stress response [5,6,7,8,9,10]. Initially, Ste20 has been shown to activate the respective MAPK cascades through Ste11 [11,12]. More recently, it has become clear that other Ste20 functions also contribute to these processes. For example, under hyperosmotic conditions, Ste20 phosphorylates histone H4 to attenuate transcription [47] and Mrc1 phosphorylation by Ste20 prevents genomic instability caused by the collision of replication and transcription machineries [48]. Since hyperosmotic stress response, pheromone response and filamentous growth are the best characterized processes of Ste20, we searched whether any of the proteins identified in the Ste20 split-ubiquitin screen are also involved in these processes. This is the case for 30 out of the 56 proteins identified in the screen (Table 3). Based on mutant phenotypes, twenty proteins play a role in filamentation, eleven in hyperosmotic stress response, and seven in pheromone response. This suggests considerable functional overlap between Ste20 and the newly identified interactors.

We also checked published genetic interactions. With the exception of *BEM1*, none of the genes identified in the split-ubiquitin screen have been reported to interact with *STE20*. *BEM1* overexpression rescues mutant phenotypes of cells lacking *STE20* [49,50]. This is not surprising since the scaffold protein Bem1 binds activated Cdc42 and several of its effectors including Ste20 and Cla4. Increased Bem1 levels could lead to the activation of a parallel pathway that shares a function with Ste20.

Negative genetic interactions between *CLA4* and genes identified in the *STE20* split-ubiquitin screen are also interesting. The deletion of either *STE20* or *CLA4* does not affect the growth rate but cells lacking both genes are inviable [51]. A negative genetic interaction between *CLA4* and another gene could mean that this gene shares a function with *CLA4* independently of *STE20* (Figure 2). Alternatively, this gene could encode a factor that acts in the same pathway as Ste20, either as upstream activator or downstream effector (Figure 2). In total, 12 out of the 56 genes identified in the *STE20* split-ubiquitin screen have previously been reported to display negative genetic interactions including synthetic lethality with *CLA4* (Table 4). Importantly, none of the genes identified in the *STE20* split-ubiquitin screen have been reported to exhibit negative genetic interactions with *STE20*, demonstrating that the genetic interactions with *CLA4* are highly specific. The combination of Ste20 protein–protein interactions and *CLA4* genetic interactions described here supports the notion that the corresponding proteins bind to Ste20 under physiological conditions and act in the same pathway as Ste20 (Figure 2).

The proteins identified in the Ste20 split-ubiquitin screen can be grouped into a relatively small number of functional categories such as translation and protein folding, glycolysis and vesicle transport (Table 1). This suggests that the proteins identified in the screen are not a random collection. Instead, the functional categories hint at potential cellular functions of Ste20, most of them having not been described previously. Several proteins could be in more than one functional group. For example, Sgt1 acts as a linker between Hsp90 and ubiquitin ligase complexes [52,53,54]. Another example is Sut1, a nuclear protein [55] that is listed here under lipid metabolism and homeostasis [56].

Taken together, we identified here 56 proteins that bind to Ste20. Only a very small number of these interactions (Bem1, Cbr1, Erg4, Ncp1, Sut1 and Vma13) have previously been characterized, leaving 50 uncharacterized interactions. Our analysis and published data, including subcellular localizations, genetic interactions, protein–protein interactions and mutant phenotypes, suggest that these interactions are probably physiologically relevant.

### 2.2. Ste20 and Glucose Metabolism

Four of the proteins identified as Ste20 interactors are glycolytic enzymes (Table 1), a quite high number considering that glycolysis is a series of only ten reactions (Figure 3). We wanted to know whether other glycolytic enzymes also bind to Ste20. Since glycolysis, gluconeogenesis and the pentose phosphate pathway are intimately linked (Figure 3) [57,58], all enzymes, major and minor isoforms [59], of these pathways were tested for interaction with Ste20. Unexpectedly, all enzymes, with the exception of the two pyruvate carboxylase isoforms Pyc1 and Pyc2, bound to Ste20 (Figure 4A). Notably, these two proteins are not specific for gluconeogenesis but are also involved in other pathways such as amino acid biosynthesis and the citric acid cycle. Despite the huge number of enzymes binding to Ste20, these interactions seem to be highly specific. Ste20 does not interact with other metabolic enzymes such as β-isopropylmalate dehydrogenase (Leu2) (Figure 4A). This enzyme catalyzes a reaction in leucine biosynthesis, a pathway that is derived from pyruvate, the end product of glycolysis. Like all glucose metabolism enzymes tested here, Leu2 also localizes to the cytoplasm [60]. Thus, Ste20 interactions seem to be limited to glucose metabolism enzymes. Furthermore, none of the glucose metabolism enzymes bound to Rdi1 (Figure 4A), demonstrating again the high specificity of these interactions. Twf1, a cytoplasmic protein involved in actin filament organization [61], which was identified in the Rdi1 split-ubiquitin screen (Appendix A), was included as another control. Twf1 does not interact with Ste20 but, as expected, binds to Rdi1 (Figure 4A). Bem1 and Cdc42 also served as controls. As expected, Cdc42 binds to both Ste20 and Rdi1, whereas Bem1 only forms a complex with Ste20 but not with Rdi1. These findings are all in line with published observations (Table 2) [15,34,35,62,63] and highlight the specificity of Ste20 interactions using this assay.

It is noteworthy that the split-ubiquitin technique not only detects direct protein–protein interactions but also picks up indirect interactions [64,65]. It is, therefore, not clear whether the observed interactions with Ste20 are direct or indirect. Nevertheless, because of the high number of enzymes binding to Ste20, we tried to confirm at least one of the interactions using an independent experimental approach. To this end, immunoprecipitation of Ste20 and Pgk1, a protein that was identified in the screen, was attempted. Ste20 co-precipitated with Pgk1 (Figure 4B), suggesting that the results of the split-ubiquitin assay reflect physiological interactions.

The pentose phosphate pathway plays a crucial role in the generation of reduced nicotinamide adenine dinucleotide phosphate (NADPH) in the cytoplasm (Figure 3) [57,58]. NADPH is required for the regeneration of reduced glutathione, one of the most important antioxidants in the cell. The synthetic compound diamide oxidizes glutathione, resulting in a decrease in the cytoplasmic pool of reduced glutathione, which causes oxidative stress [66,67]. It is, therefore, not surprising that changing the activity of glucose metabolism enzymes that can alter NADPH levels has an effect on diamide sensitivity. For example, cells with reduced catalytic activity of the glycolytic enzyme triosephosphate isomerase (Tpi1) accumulate NADPH which confers diamide resistance [68,69]. Since Ste20 binds to glucose metabolism enzymes, it is tempting to speculate that Ste20 modulates the activity of these enzymes. It was, therefore, tested whether altering *STE20* levels affects diamide sensitivity. *STE20* overexpression resulted in an increase in diamide sensitivity (Figure 5A), whereas cells lacking *STE20* exhibited diamide resistance (Figure 5B). This suggests that Ste20 negatively modulates NADPH levels possibly through the regulation of glucose metabolism enzymes.
Figure 3Overview of glycolysis, gluconeogenesis and the pentose phosphate pathway. In glycolysis (left, red arrows), glucose is oxidized to pyruvate. In gluconeogenesis (left, red arrows), pyruvate is converted to glucose 6-phosphate. Most of the gluconeogenic enzymes are also used in glycolysis. Gluconeogenesis-specific enzymes are Pyc1, Pyc2, Pck1 and Fbp1. Unlike some higher eukaryotes, budding yeast has no glucose 6-phosphatase. In contrast to many higher eukaryotes, the enzymes catalyzing the conversion of pyruvate to oxaloacetate (Pyc1 and Pyc2) are not mitochondrial but cytoplasmic [70]. In the oxidative branch of the pentose phosphate pathway (right, green arrows) glucose 6-phosphate is converted into ribulose 5-phosphate. This part includes two reactions that generate NADPH. In the non-oxidative branch of the pentose phosphate pathway (right, black arrows), pentoses such as ribulose 5-phosphate and the glycolytic/gluconeogenic intermediates fructose 6-phosphate and glyceraldehyde 3-phosphate are interconverted. Some reactions are catalyzed by multiple redundant isoenzymes. In contrast, Pfk1 and Pfk2 are two distinct subunits of the hetero-oligomeric phosphofructokinase [71]. For simplification, ADP/ATP and NAD^+^/NADH are not shown. Single-headed arrows designate irreversible reactions, and double-headed arrows indicate reversible reactions. Proteins that were identified in the Ste20 split-ubiquitin screen are shown in green, all other proteins in blue.
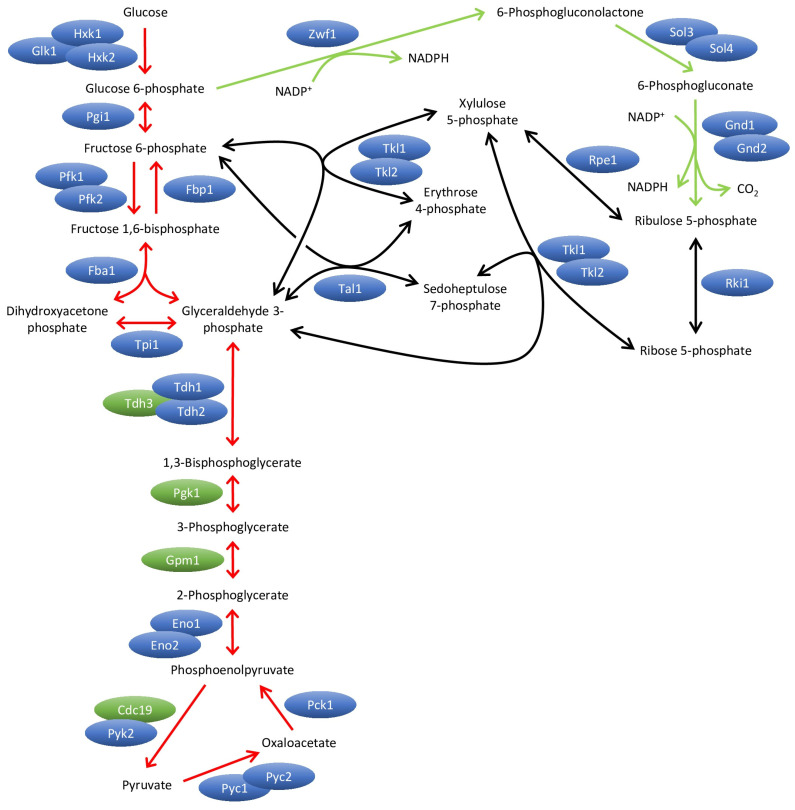

Figure 4Ste20 binds to glucose metabolism enzymes. (**A**) Ste20 interacts with enzymes of glucose metabolism using the split-ubiquitin assay. Here, 10^4^ cells of the indicated plasmid combinations were spotted onto plates lacking histidine and leucine (+uracil) to select for the plasmids, and onto plates lacking histidine, leucine and uracil (−uracil) to monitor protein–protein interactions. Ste20 and the control protein Rdi1 are both fused to the C-terminal half of ubiquitin and the reporter RUra3 (Figure 1). The other proteins are fused to the N-terminal half of ubiquitin. Leu2, Twf1, Bem1 and Cdc42 were included as controls for Ste20 and Rdi1. (**B**) Co-immunoprecipitation of Ste20 and Pgk1. Cells expressing *STE20*-*3HA*, and *STE20*-*3HA* with *PGK1*-*9myc* were lysed and equal amounts of protein were precipitated with anti-myc antibodies. Immunoblots were probed with antibodies raised against the myc and HA epitope.
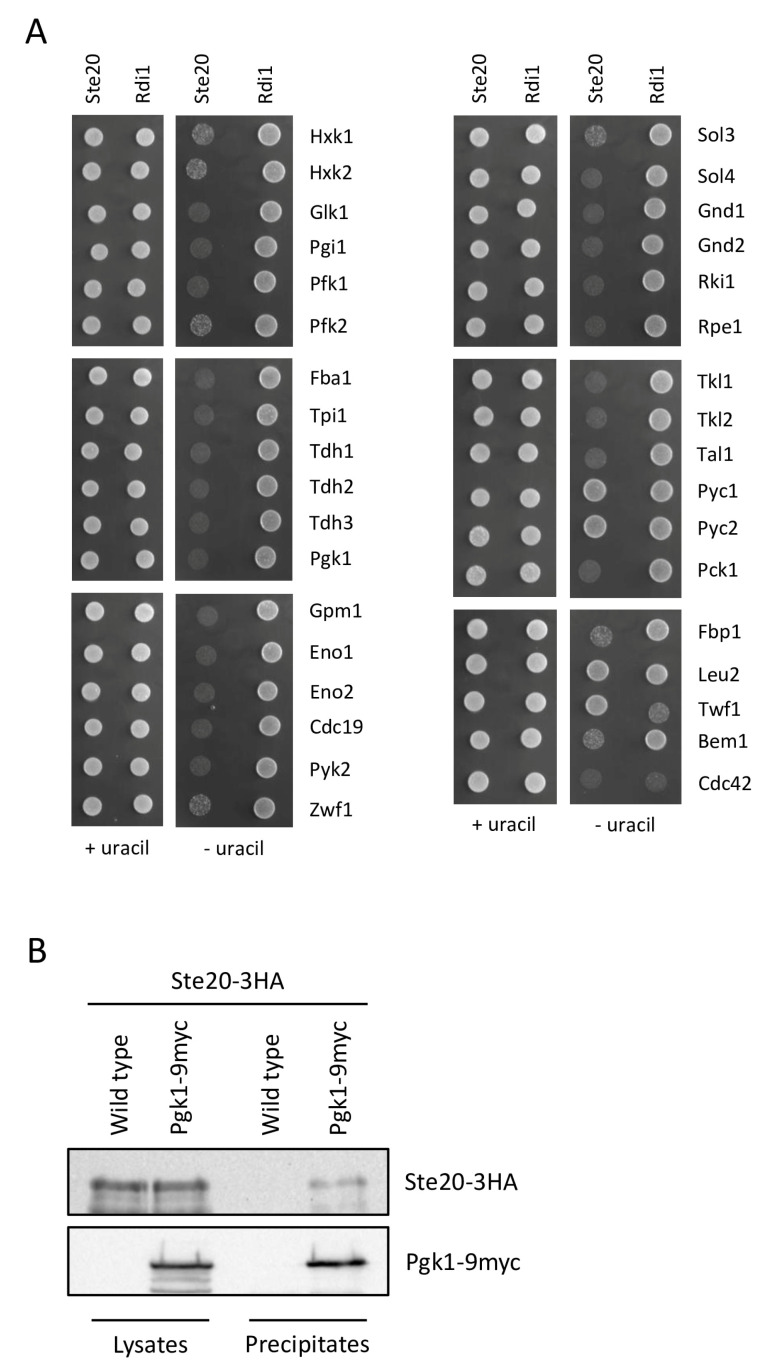

Figure 5*STE20* levels affect diamide sensitivity. (**A**) Overexpression of *STE20* increases diamide sensitivity. Serial dilutions (1:10) of the indicated strains were spotted on selective medium containing either no diamide or diamide at a concentration that is sublethal for the wild type (2 mM). The *STE20* overexpression strain carried the multicopy plasmid pRS426 with *STE20* under control of its endogenous promoter. The wild type carried the empty pRS426 plasmid. (**B**) *STE20* deletion confers diamide resistance. Serial dilutions (1:10) of the indicated strains were spotted on YPD medium containing either no diamide or diamide at a concentration that is lethal for the wild type (3 mM).
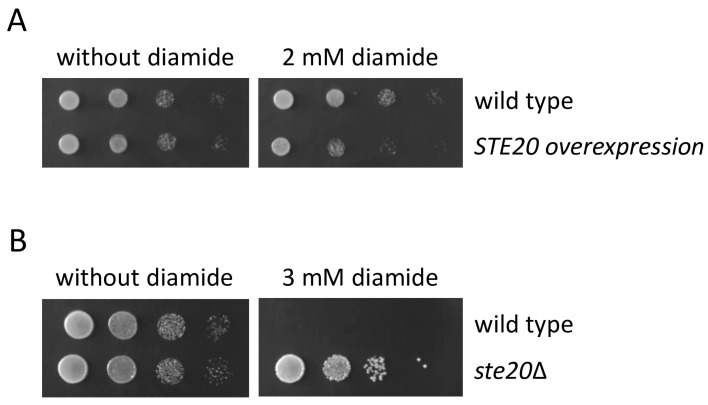



In summary, Ste20 binds to a large number of glucose metabolism enzymes either directly or indirectly, and these interactions are probably physiologically relevant.

### 2.3. Interactions between Ste20 and Nuclear Proteins

Ste20 can translocate to the nucleus where it has a range of functions, including the regulation of apoptosis through histone H2B [13], the attenuation of stress gene expression via histone H4 [47] and the modulation of the transcription of sterol uptake genes through the transcriptional regulator Sut1 [17]. Using the split-ubiquitin screen, we identified another nine nuclear proteins as Ste20 interactors (Table 1). In this study, we focus on three key regulators of various steps of gene expression: Sac3, Hmt1 and Ctk1.

Sac3 is a component of the transcription export complex 2 (TREX-2). This protein complex plays a central role in the integration of the nuclear export of mature mRNA with earlier steps in the gene expression pathway including transcription and RNA processing [72]. Sac3 serves as a scaffold that binds the other TREX-2 components, Thp1, Sus1, Sem1 and Cdc31 [73,74,75,76] (Figure 6A). The binding sites of Sac3 for mRNA, the principal mRNA export factors Mex67-Mtr2, the TREX-2 components Thp1, Sem1, Sus1 and Cdc31, the nuclear pore complexes and the Mediator complex, an essential regulator of RNA polymerase II, can all be found in Sac3 residues ~1–805 (Figure 6A). In contrast, no proteins are known to interact with the C-terminal residues ~806–1301, and no function has been attributed to this region. It is noteworthy, that in our split-ubiquitin screen, the full-length Sac3 was not identified as a Ste20 interactor, but a C-terminal Sac3 fragment comprising only residues 968–1301. However, the full-length Sac3 also binds to Ste20 (Figure 6B). Since the C-terminal 334 residues of Sac3 are sufficient for interaction with Ste20, more Sac3 fragments were generated to further narrow down the Ste20-binding site (Figure 6A). Ste20 interacted with truncated Sac3 comprising C-terminal 400, 300, 200 and 100 residues (Figure 6B). However, no binding was observed for a Sac3 fragment of the last 50 residues (Figure 6B), indicating that the 100 C-terminal Sac3 residues are sufficient and necessary for the Ste20 interaction.

Ste20 lacking its nuclear localization signal (NLS) (Ste20^∆NLS^) is functional but it no longer translocates to the nucleus [17]. The full-length Sac3 did not bind to Ste20^∆NLS^ (Figure 6B), suggesting that the Ste20–Sac3 interaction takes place in the nucleus. Ste20 that lacks its NLS still binds to Bem1 (Figure 6B), an interaction that occurs at the plasma membrane [26,28], demonstrating that Ste20^∆NLS^ is suitable for monitoring protein–protein interactions outside the nucleus. All C-terminal fragments of Sac3, apart from the smallest one, also bound to Ste20^∆NLS^ (Figure 6B). This is not unexpected because a Sac3 NLS has been predicted for residues 701–717 [81] (Figure 6A). The truncated Sac3 versions lack this putative NLS and, therefore, presumably can be found in the cytoplasm where they could interact with Ste20^∆NLS^. Furthermore, Ste20 bound to the other TREX-2 subunits (Thp1, Sem1, Sus1 and Cdc31) (Figure 6B). These interactions all required the Ste20 NLS (Figure 6B).

For these experiments, Cla4 was included as a control because it is a PAK related to Ste20 [51], and like Ste20, Cla4 localizes not only to the cytoplasm and plasma membrane but also the nucleus [17,82]. Cla4 bound to the control protein Bem1, confirming a well-established link [30], but it did not interact with the full-length Sac3, any of the Sac3 fragments and the other TREX-2 components (Figure 6B). This again demonstrates the high specificity of split-ubiquitin interactions.

Several large-scale screens revealed negative genetic interactions including synthetic lethality between *CLA4* and *SAC3* [83,84,85,86], *SUS1* [84], *SEM1* [83,84,85,86,87,88,89,90] and *THP1* [84] deletions. These interactions were confirmed here (Figure 6C). In our hands, the deletion of either *SAC3*, *SEM1*, *SUS1* or *THP1* did not affect the growth rate of these strains (Figure 6C). However, in all cases, we observed synthetic lethality when combined with the deletion of *CLA4* deletion (Figure 6C). In contrast, *STE20* deletion has no effect on the growth of cells also lacking *SAC3*, *SUS1*, *SEM1* or *THP1* (Figure 6C). As mentioned above (Figure 2), such a high specificity of genetic and protein–protein interactions suggests that Sac3 and the other TREX-2 components act in the same pathway as Ste20, possibly as downstream effectors of Ste20.

The loss of *SAC3* results in a strong nuclear accumulation of mRNA, demonstrating a crucial role for Sac3 in nuclear mRNA export [73,91]. Therefore, it would seem reasonable to expect numerous serious defects in cells lacking *SAC3*. However, under optimal lab conditions, the *SAC3* deletion strain grows like the wild type or only displays slightly reduced growth (Figure 6C and Figure 7A,C), and very few other obvious defects have been reported. In this study, we found that Sac3, like Ste20, has a role in filamentous growth, a differentiation process that is triggered by nutrient depletion. Filamentous growth can be observed in both haploids and diploids, but the stimuli that lead to it, the underlying signaling pathways and the morphological responses differ slightly in these cell types [92]. In haploid cells, the lack of a fermentable carbon source such as glucose results in an invasion of the agar substratum. This form of filamentous growth is, therefore, also known as invasive growth. Diploids grown under nitrogen limitation elongate and move away from the colony, a process that is termed pseudohyphal growth. In the homozygous *SAC3* deletion strain, diploid pseudohyphal growth is completely absent (Figure 7A), and cells overexpressing *SAC3* display increased pseudohyphal growth (Figure 7B). In contrast, in haploid cells, *SAC3* deletion or overexpression does not affect invasive growth (Figure 7C,D). These distinct phenotypes have also been observed for the *ste20*^∆*NLS*^ mutant. Wild type *STE20* is essential for both haploid invasive and diploid pseudohyphal growth [7,8] (Figure 7A,C). Cells expressing *STE20* without its NLS exhibit normal haploid invasive growth (Figure 7C) as previously shown [17]. Here, we found that, unlike the *STE20* wild type allele, *STE20*^∆*NLS*^ does not complement the diploid pseudohyphal growth defect of the homozygous *STE20* deletion strain (Figure 7E). An overexpression of wild type *STE20* and *STE20*^∆*NLS*^ had no effect on filamentous growth in either haploid or diploid cells (Figure 7B,D). Taken together, this suggests that Ste20 has an essential nuclear function in diploid pseudohyphal growth but not in haploid invasive growth.

Next, we examined Hmt1, the predominant arginine methyltransferase in budding yeast [93]. Through the methylation of proteins such as Hrp1, Nab2, Npl3, Snp1, Yra1 and histone H4, Hmt1 regulates a wide range of processes in gene expression. This includes transcription elongation and termination, the splicing of pre-mRNA, the nuclear export of mRNA and the formation of silent chromatin [94,95,96,97,98,99].

Hmt1 binds to Ste20, an interaction that requires the nuclear localization of Ste20 (Figure 6B). The related PAK Cla4 does not associate with Hmt1 (Figure 6B). Like the *ste20*^∆*NLS*^ mutant and the *SAC3* deletion strain, *hmt1*∆ cells display normal haploid invasive growth (Figure 7C), and diploid pseudohyphal growth is completely absent in *hmt1*∆/*hmt1*∆ cells (Figure 7A). Furthermore, *HMT1* overexpression results in a strong increase in diploid pseudohyphal growth (Figure 7B) but has no effect in haploid cells (Figure 7D).

Finally, we examined the link between Ctk1 and Ste20. Ctk1 is the catalytic subunit of the heterotrimeric C-terminal domain (CTD) kinase I complex [100]. It phosphorylates the CTD of Rpo21 (also known as Rpb1), the largest RNA polymerase II subunit. This CTD consists of tandem heptapeptide repeats whose dynamic phosphorylation coordinates transcription with co-transcriptional events [101]. For example, CTD phosphorylation by Ctk1 plays an important role in pre-mRNA 3′ processing [102,103].

Like the other nuclear interactors of Ste20 tested here, the binding of Ste20 to Ctk1 requires the nuclear localization of Ste20, and Ctk1 does not interact with Cla4 (Figure 6B). Since *CTK1* is essential in the Σ1278b background [104], which is used for filamentous growth assays, the effect of *CTK1* deletion on invasive growth and pseudohyphal growth could not be tested. As for *SAC3* and *HMT1*, the overexpression of *CTK1* increased diploid pseudohyphal growth (Figure 7B) but had no effect on haploid invasive growth (Figure 7D).

As reported previously, cells lacking *CTK1* grew extremely slowly (Figure 8A) [100]. The deletion of *CLA4* in the *ctk1*∆ strain had no effect on the growth rate but, unexpectedly, *STE20* deletion rescued the severe growth defect of *ctk1*∆ cells (Figure 8A). When a copy of wild type *STE20* was brought back to the *ctk1*∆ *ste20*∆ double mutant, growth was again greatly reduced as expected, but expression of *STE20*^∆*NLS*^ had no effect in the *CTK1 STE20* double deletion strain (Figure 8A). These observations suggest that Ste20 and Ctk1 act in the same process in the nucleus, possibly the modification of RNA polymerase II CTD. Our observations can be explained by a model in which Ste20 functions as an inhibitor of an unknown protein that, like Ctk1, modifies the CTD (Figure 8B). However, compared with Ctk1, CTD modification by this hypothetical protein is less important. Since Ctk1 plays such a crucial role in CTD phosphorylation, cells lacking *CTK1* display a severe growth defect but are still viable due to activity of other kinases including the hypothetical protein. Due to the lack of Ste20 inhibition in a *STE20* deletion strain, the hypothetical protein would be more active but this would not affect cell growth. However, in the *ctk1*∆ *ste20*∆ double mutant, increased activity of the unknown protein could compensate for the absence of *CTK1*. This model would suggest that *STE20* overexpression could inhibit the unknown protein even more (Figure 8B). In the presence of *CTK1*, reduced activity of the hypothetical protein probably would not affect the growth rate. However, in a *ctk1*∆ strain, *STE20* overexpression could exacerbate the growth defect of *ctk1*∆. To test whether this is the case, we overexpressed *STE20* and *STE20*^∆*NLS*^ using the strong, inducible *GAL1* promoter. Increased *STE20* levels did not affect the growth of wild type cells (Figure 8C). In the *ctk1*∆ strain, *STE20* overexpression reduced the growth even further. Notably, the overexpression of *STE20* lacking its NLS had no effect (Figure 8C). These observations are in line with the proposed model, suggesting that Ste20 has a nuclear function acting in parallel with Ctk1.

In summary, Ste20 can translocate to the nucleus where it binds with high specificity to Sac3, Hmt1 and Ctk1. These proteins are key regulators integrating various aspects of gene expression from transcription to nuclear mRNA export. Nuclear Ste20, Sac3, Hmt1 and Ctk1 all play important roles in diploid pseudohyphal growth. This functional overlap and physical, as well as genetic, interactions suggest that nuclear Ste20 acts in the same processes as Sac3, Hmt1 and Ctk1.

### 2.4. Other Interactions

The ribosomal protein Rpl13B and seven other proteins involved in translation and protein folding were identified in the split-ubiquitin screen as Ste20 interactors (Table 1). This was further analyzed to rule out that these interactions are just artefacts that occur when Ste20 is synthesized at ribosomes. Rpl13B, a protein of the large ribosomal subunit, and the ribosome-associated chaperones Ssb2 and Zuo1 all bound specifically to Ste20 but not to the control protein Rdi1 (Figure 9A). Rpl13A, the highly similar paralog of Rpl13B, which was included here as another control, did not interact with either Ste20 or Rdi1.

*CLA4* deletion has been reported to be synthetically lethal with the deletion of seven genes encoding ribosomal subunits (*RPL17A*, *RPL19A*, *RPL23A*, *RPL24A*, *RPL35A*, *RPS21A*, *RPS21B*) [83,87,88]. In contrast, no genetic interactions between *STE20* and any genes encoding ribosomal proteins have been published. Because of the high number of known genetic interactions between *CLA4* and ribosomal genes, we also analyzed *RPL13B* and its paralog *RPL13A*. The deletion of *RPL13B* results in a severe growth defect, whereas *rpl13a*∆ cells grow like the wild type (Figure 9B). *STE20* deletion in these strains does not alter the growth rate. In contrast, the *rpl13b*∆ *cla4*∆ double deletion strain is not viable (Figure 9B). This synthetic lethal interaction is highly specific because *rpl13a*∆ *cla4*∆ cells are indistinguishable from the wild type.

Another group of Ste20 interactors identified in the split-ubiquitin screen comprises five proteins with ubiquitin-related functions (Table 1). Since the screen utilizes ubiquitin and ubiquitin-specific proteases (Figure 1), it was important to find out whether the identified proteins were simply artefacts. Ubp3, a ubiquitin-specific protease [105], and Ubx7, a ubiquitin regulatory X (UBX) domain-containing protein involved in ubiquitin-dependent protein degradation [106,107], both bound to Ste20 only and not to the control protein Rdi1. This suggests that these interactions are indeed specific and independent of the experimental approach employing ubiquitin (Figure 9A).

Taken together, the high specificity of protein–protein and genetic interactions argues against the notion that the interactors identified in the Ste20 split-ubiquitin screen merely represent artefacts. Rather, the interactions seem physiological with a potential role for Ste20 modulating translation and protein folding as well as ubiquitin-dependent protein degradation.

## 3. Discussion

Since PAKs were first described around 30 years ago [108], considerable progress has been made in the understanding of these key signaling proteins [1,2,3,4,21,22]. However, a lot of questions still remain unanswered. In order to get a comprehensive insight into the functions and the integration of various cellular activities of a PAK, we screened for the binding partners of the budding yeast PAK Ste20 as an example. Using the split-ubiquitin technique, we identified 56 proteins that physically interact with Ste20 in this study (Table 1). Most of them have not been reported to bind to Ste20 before. This raises the question whether these are physiologically relevant interactions. We believe this is the case for several reasons. (1) Some of the interactors have independently been identified by others (Table 2). (2) Interactions between Ste20 and five proteins (Cbr1, Erg4, Ncp1, Sut1 and Vma13) described in this screen have previously been characterized by us [14,17,32,33]. (3) The split-ubiquitin interactions described here are highly specific. Another split-ubiquitin screen using the same library and Rdi1 as bait identified 50 proteins (Appendix A). Out of these, only Mlc1 was also found in both screens. Other interactions presented in this study also demonstrate the high specificity of the split-ubiquitin technique. For example, Ste20 only binds to the ribosomal protein Rpl13B but not to the highly similar Rpl13A (Figure 9A). (4) All of the interactors listed here, can be found in subcellular locations where they would be able to physically interact with Ste20. (5) Ste20 has a well characterized role in the activation of MAPK pathways mediating hyperosmotic stress response, pheromone response and filamentous growth [5,6,7,8,9,10]. Previously reported mutant phenotypes indicate a role in these processes for 30 out of the 56 proteins identified here as Ste20 interactors (Table 3). In addition, we show in this study that Sac3, Hmt1 and Ctk1 also have important roles in filamentous growth. This considerable functional overlap is consistent with physical interactions between Ste20 and these proteins. (6) The simultaneous deletion of *STE20* and *CLA4* is lethal [51], suggesting that Ste20 has one or more functions that are essential in cells lacking *CLA4*. Likewise, the deletion of genes-encoding proteins that act in the same pathway as Ste20 and share a function with Cla4 can be lethal in a *cla4*Δ background [87]. For 12 genes identified in the Ste20 split-ubiquitin screen, negative genetic interactions with a *CLA4* deletion have previously been published (Table 4). We also demonstrated here the synthetic lethality for the *rpl13b*∆ *cla4*∆ strain but not for the related *rpl13a*∆ *cla4*∆ double mutant (Figure 9B). (7) The proteins identified in the screen fall into a relatively small number of functional groups such as vesicle transport and glycolysis (Table 1), suggesting that these proteins are not random hits and that Ste20 might be involved in these processes.

According to the *Saccharomyces* Genome Database, 155 proteins have been described by others to physically interact with Ste20 (Table 5). Most of these interactions have been found through high-throughput screens and have not been further characterized. In some cases, the Ste20 interactors described in this study and proteins identified by others are quite similar. For example, we have found the ubiquitin-specific protease Ubp2 and the UBX domain-containing Ubx7, while others have identified two more ubiquitin-specific proteases (Ubp7 and Ubp9) and another UBX domain-containing protein (Ubx5) [109].

Like the Ste20 interactors identified by us, the huge majority of the proteins that physically interact with Ste20 described by others can also easily be grouped together according to their functions (Table 5). The functional categories for the Ste20 interactors described by us and by other groups show considerable overlap (Table 1 and Table 5). Some of these functions include well established roles for Ste20 such as MAPK signaling and cell polarity [4,110]. However, other functions, such as ribosome biogenesis and tRNA synthesis and modification, are rather unexpected and these new links are certainly worth examination in the future. Notably, some of the functional categories are connected and/or overlap. For example, vesicle transport plays an important role in cell polarity [110]. Another observation worth noting is that around half the Ste20 interactors published by others are involved in various aspects of gene expression (Table 5). The potential roles for Ste20 in the regulation of gene expression will be discussed below. To summarize, the large number of Ste20 interactors identified by us and many other groups falling into a quite small number of functional categories suggests the possible involvement of Ste20 in these processes.

We examined the link between Ste20 and glucose metabolism in a bit more detail. Here, we show that Ste20 interacts with most glucose metabolism enzymes tested (Figure 4A). Importantly, the observed protein–protein interactions are not necessarily direct as the split-ubiquitin assay picks up both direct and indirect interactions [64,65]. Even for a kinase with a large number of substrates, an association with all glucose metabolism enzymes seems unlikely. In budding yeast, glycolytic enzymes form a multi-enzyme complex which binds to actin filaments [111]. Similar glycolytic metabolons have also been described for mammalian cells [112,113]. Ste20 could, therefore, interact with only a few of the enzymes directly. The other proteins would just be part of the same protein complex. The confirmation of one of the interactions (Ste20-Pgk1) by co-immunoprecipitation (Figure 4B), the fact that Ste20 does not bind to an unrelated metabolic enzyme (Leu2) from a pathway that is linked to glucose metabolism and that is also present in the cytoplasm [60] (Figure 4A) and the fact that the control protein Rdi1 does not bind to any of the enzymes (Figure 4A) all suggest that the interactions observed here are indeed specific and occur under physiological conditions.

It is conceivable that Ste20 not only binds to these enzymes but that it could modulate glucose metabolism. Diamide resistance in the *STE20* deletion strain and increased diamide sensitivity in cells overexpressing *STE20* (Figure 5A,B) suggest that Ste20 has a negative impact on NADPH generation probably through the modulation of the pentose phosphate pathway. The regulation of glucose metabolism by PAKs has been described before. PAK1, the closest mammalian homologue of Ste20, interacts with two glucose metabolism enzymes, phosphoglucomutase and phosphoglycerate mutase [114,115,116]. Phosphoglucomutase catalyzes the interconversion of glucose 6-phosphate and glucose 1-phosphate, a reaction that links several pathways, for example, glycolysis/gluconeogenesis and the pentose phosphate pathway to glycogen synthesis and breakdown. The phosphorylation of phosphoglucomutase by PAK1 increases the activity of this enzyme [115]. PAK1 also phosphorylates phosphoglycerate mutase, which catalyzes the interconversion of 3-phosphoglycerate and 2-phosphoglycerate [114,116]. This phosphorylation leads to the ubiquitin-mediated degradation of the enzyme.

Interestingly, PAK1 not only modulates glucose metabolism, but it also plays a central role in other aspects of glucose homeostasis [117]. To name a few examples, PAK1 is involved in insulin secretion in pancreatic β-cells, insulin-dependent glucose uptake in skeletal muscle cells and in the production of the incretin hormone glucagon-like peptide-1 (GLP-1) in intestinal L cells [117]. These examples clearly illustrate that PAKs not only act on a few substrates but that these kinases can regulate complex biological processes such as glucose homeostasis at many different levels.

Sac3, Ctk1 and Hmt1 all play important roles in diploid pseudohyphal growth but not in haploid invasive growth (Figure 7A–D). Ste20 is essential for both pathways, which is mediated via MAPK activation at the plasma membrane [7,8,11]. We now show that the nuclear translocation of Ste20 is also required, but only for diploid pseudohyphal growth (Figure 7C,E) [17]. This suggests that Ste20, in addition to its MAPK activation, also has one or more nuclear functions specifically relevant for diploid pseudohyphal growth. Since Ste20 physically interacts with Sac3, Ctk1 and Hmt1, and because of similar phenotypes, this raises the interesting possibility that during diploid pseudohyphal growth, Ste20 could modulate the activity of Sac3, Ctk1 and Hmt1. Pseudohyphal growth is a stress response to nutrient depletion. To cope with extreme stress, cells usually instantly synthesize proteins that mediate a stress response. This is achieved by preferentially transcribing, processing, exporting and translating mRNAs that encode the proteins needed for adaptation to stress [118]. It would, therefore, make sense that in response to nutrient depletion, Ste20 modulated the activity of Sac3, Hmt1 and Ctk1 not constitutively for all mRNAs but only for the subset of transcripts that are required for pseudohyphal growth. Selective mRNA export has been reported for germinal center-associated nuclear protein (GANP), the human homologue of Sac3 [119,120]. GANP exports a subset of mRNA involved in stress response from the nucleus at a higher rate. This presumably facilitates a rapid adaptation to a change in environmental conditions. Transcript specificity has also been described for Hmt1 [121]. The mitotic exit kinase Dbf2 phosphorylates and thereby activates Hmt1 which results in the transcript-specific stabilization of cyclin *CLB2* mRNA. This leads to an accumulation of Clb2 protein which then promotes entry into mitosis.

What does all this mean? A wide range of functions has been attributed to Ste20; many of them do not seem to have much in common. By combining the results presented here and data published by others, we believe that most of the Ste20 functions are not unrelated but actually complement each other. For example, a major biological function that involves Ste20 is the response to extracellular signals such as pheromones, high solute concentrations and low nutrient levels. These signals trigger the three distinct MAPK pathways mentioned above. The stimulation of these pathways by Ste20 at the plasma membrane eventually leads to transcription factor activation. To ensure an efficient gene expression, Ste20 also has a range of other functions that are independent of MAPK stimulation. This includes Ste20 phosphorylation of Mrc1, a regulatory component of the DNA replication complex [48]. This leads to the inhibition of DNA replication and avoids collision of transcription and replication machineries. This is a critical process because these collisions are a major source of genomic instability. In response to hyperosmotic stress, Ste20 also phosphorylates histone H4 to attenuate the transcription of stress genes [47]. This ensures an optimal duration of the stress response which is crucial for cell survival. As mentioned above, we suggest that Ste20 regulates efficient and selective transcription, pre-mRNA processing and nuclear mRNA export trough Ctk1, Hmt1 and Sac3. In response to nutrient deprivation and other stress, Ste20 also modulates selective mRNA degradation via the phosphorylation of Dcp2, the catalytic subunit of the mRNA decapping enzyme complex [122]. Finally, due to the large number of proteins involved in translation that physically interact with Ste20 (Table 5), it seems likely that Ste20 also plays a role in protein biosynthesis, possibly modulating translational activity. Again, this could be a transcript-specific process.

The association of Ste20 with mRNA and mRNA-binding proteins is well documented. Ste20 binds to mRNA when nutrients are limited [123], which could be relevant for inducing filamentous growth. In addition, at least 15 mRNA-binding proteins interact with Ste20 (Table 5). Some of these mRNA-binding proteins (Gsp1, Rim4, She3, Tho1) are phosphorylated by Ste20 [109], and other proteins (Ded1, Pat1, Rie1, Sbp1) are phosphorylated in a Ste20-dependent manner during pseudohyphal growth [124]. Furthermore, several mRNA-binding proteins are required for pseudohyphal growth (Ccr4, Dhh1, Lsm1, Pat1, Pbp1, Sbp1) [124]. These observations further support the idea that Ste20 could play a key role in the regulation of gene expression, in particular during pseudohyphal growth.

It is quite remarkable that Ste20 could act at so many levels of gene expression. However, this is not really surprising since this would ensure an efficient adaptation to changing environmental conditions which is crucial for cell survival. In conclusion, based on our results and published data, we suggest that Ste20 might have a range of new and hitherto unappreciated functions, in particular in all aspects of gene expression.

The results obtained in budding yeast could also be relevant for higher eukaryotes. Some of the Ste20 interactors identified here have already been shown to bind to mammalian PAK1. As mentioned above, PAK1 phosphorylates phosphoglycerate mutase [114,116]. Another example is the myosin light chain Mlc1. In higher eukaryotes, MLC is phosphorylated by PAK1 [125,126,127].

## 4. Materials and Methods

### 4.1. Yeast Strains, Plasmids and Growth Conditions

All yeast strains used in this study are listed in Table 6. The strains were in the YPH499 background, with the exception of strains used for haploid invasive growth and diploid pseudohyphal growth which were in the Σ1278b background. Yeast strains were constructed using PCR-amplified cassettes [128,129,130]. The strains were grown in standard rich medium containing 1% yeast extract, 2% peptone, 2% dextrose (YPD) or minimal synthetic complete (SC) medium. For induction of the *GAL1* promoter, yeast cells were grown in medium with 2% galactose and 3% raffinose instead of glucose. Media for the split-ubiquitin screen and loss of centromeric *URA3*-containing plasmids were supplemented with 1 mg/mL 5-FOA and 50 µg/mL uracil. All constructs used in this work are listed in Table 7.

### 4.2. Split-Ubiquitin Interaction Analysis

For the split-ubiquitin screens, *ste20*∆ cells carrying P*_MET25_*-*STE20*-*CUBI*-*RURA3* and *rdi1*∆ cells carrying P*_MET25_*-*RDI1*-*CUBI*-*RURA3* were transformed with a split-ubiquitin library (a gift from Nils Johnsson, Ulm University, Germany) and grown on 5-FOA plates for 3 days at 30 °C. Genes were subcloned after PCR amplification to confirm the interaction. Plates lacked cysteine and methionine for expression of *STE20* and *RDI1* fusions that were under the control of the *MET25* promoter. Plates also lacked histidine and leucine to select for the presence of fusion constructs.

### 4.3. Immunoprecipitation and Immunoblotting

Next, 100 mL cells in the exponential phase were grown until they reached an optical density of 1 (measured at 600 nm). Cells were harvested and washed with water. After washing with lysis buffer (20 mM Tris, pH 7.5, 100 mM sodium chloride, 10 mM EDTA, 1 mM EGTA, 5% glycerol, 1% Triton X-100), cells were disrupted with glass beads in lysis buffer and clarified using centrifugation at 13,000 rpm for 5 min. Protein concentration was determined using Bradford protein assay solution. Pgk1-9myc was immunoprecipitated by adding mouse monoclonal anti-myc antibody (9E10; Santa Cruz Biotechnology) and protein G-Sepharose (GE Healthcare). Resin was washed three times with lysis buffer, resuspended in 2× SDS sample buffer, and analyzed using immunoblotting. Mouse monoclonal anti-HA (12CA5) was obtained from Santa Cruz Biotechnology, and secondary antibodies were from Jackson ImmunoResearch Laboratories.

### 4.4. Filamentation Assays

For haploid invasive growth assays, 10^5^ cells of an overnight culture were spotted on YPD or selective medium, and grown at 30 °C. Plates were photographed before and after being rinsed under a stream of deionized water. For pseudohyphal growth assays, cells were grown over night, and 100 cells were spread on solid synthetic low ammonium dextrose (SLAD) medium containing 0.67% yeast nitrogen base without amino acids and without ammonium, 2% glucose and 50 μM ammonium sulfate. Plates were incubated at 30 °C. Colonies were examined with a Zeiss Axioplan 2 microscope equipped with a 5× objective and images were captured using a ProgRes C12 Plus camera (Jenoptik, Jena, Germany).

## Figures and Tables

**Figure 1 ijms-24-15916-f001:**
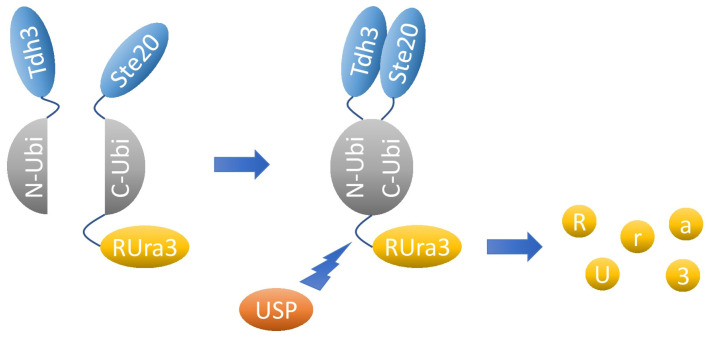
The split-ubiquitin technique. The N-terminal and C-terminal halves of ubiquitin (N-Ubi and C-Ubi) alone do not assemble. If the glycolytic enzyme Tdh3, shown here as an example, linked to N-Ubi binds to Ste20, which is fused to C-Ubi, both ubiquitin halves can form a quasi-native ubiquitin. This is recognized by a ubiquitin-specific protease (USP) which cleaves off the reporter RUra3 that is fused to C-Ubi-Ste20. The released RUra3, a modified version of the enzyme Ura3 with an arginine at the extreme N-terminus, is rapidly degraded by the N-end rule pathway. Interaction between Tdh3 and Ste20, therefore, results in uracil auxotrophy. Since Ura3 converts 5-fluoroorotic acid (5-FOA) into the toxic 5-fluorouracil, the binding of Tdh3 to Ste20 also results in 5-FOA resistance.

**Figure 2 ijms-24-15916-f002:**
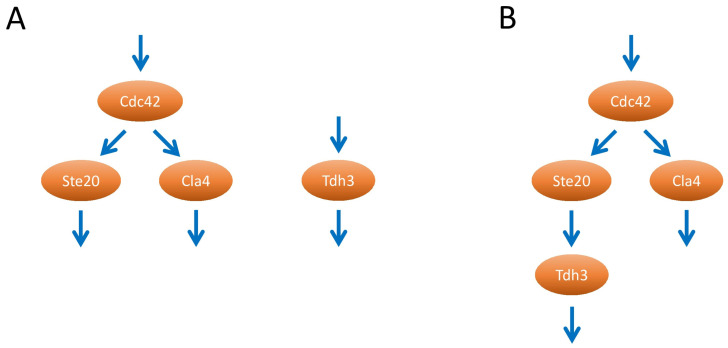
Interpretation of negative genetic interactions with *CLA4*. The Rho GTPase Cdc42 activates several downstream effectors, including the related p21-activated kinases (PAKs) Ste20 and Cla4. *TDH3*, which encodes a glycolytic enzyme, is shown here as an example for a gene that displays a negative genetic interaction with *CLA4*. (**A**) The Tdh3 protein could share a function with Cla4 and act in a pathway independently of Ste20. (**B**) However, since Tdh3 also physically interacts with Ste20 it is likely to act in the same pathway as Ste20, for example, as shown here, as a downstream effector.

**Figure 6 ijms-24-15916-f006:**
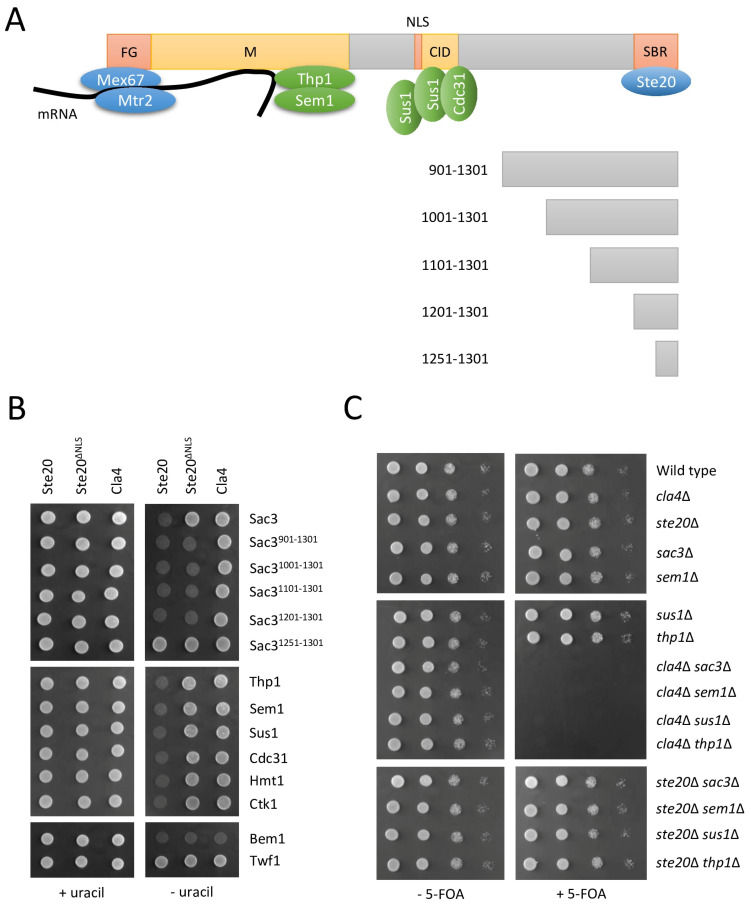
Ste20 interacts with the transcription export complex 2 (TREX-2) scaffold protein Sac3. (**A**) Domain organization in Sac3. The protein is 1301 residues long. N-terminal phenylalanine-glycine (FG) repeatedly binds to the heterodimeric principal mRNA export factor Mex67-Mtr2 which associates with mRNA [77]. The middle (M) region of Sac3 forms a complex with Thp1 and Sem1, creating an mRNA docking site [78,79]. The Cdc31-interacting domain (CID) region binds to Cdc31 and two copies of Sus1 [80]. Also included are a predicted bipartite nuclear localization signal (NLS) [81] and the Ste20-binding region (SBR) identified in this study. Not shown for simplification are the interactions between the CID region and nuclear pore complexes, and between the M-region and the Mediator complex, a key regulator of RNA polymerase II. TREX-2 components that bind to Sac3 are shown in green, and other proteins that interact with Sac3 are shown in blue. The lower panel presents the C-terminal Sac3 fragments used to determine the Ste20-binding site. Sac3 domain boundaries and Sac3 fragments are drawn to scale. (**B**) Split-ubiquitin interactions between Ste20 and nuclear proteins. Strains carrying the indicated plasmids were grown on selective medium supplemented with uracil as control, and on selective medium lacking uracil to detect protein–protein interactions. Ste20^ΔNLS^ lacks its NLS (residues 272–288) [17]. Bem1 and Twf1 were included as positive and negative controls, respectively. (**C**) Genetic interactions between the PAKs *STE20* and *CLA4*, and genes encoding TREX-2 subunits. Serial dilutions (1:10) of the indicated strains were spotted on selective medium with or without 5-FOA. Strains lacking *CLA4* carried a *CLA4* copy on a pRS316 plasmid. All other strains carried an empty pRS316 vector. 5-FOA selects against the *URA3*-based pRS316 plasmids.

**Figure 7 ijms-24-15916-f007:**
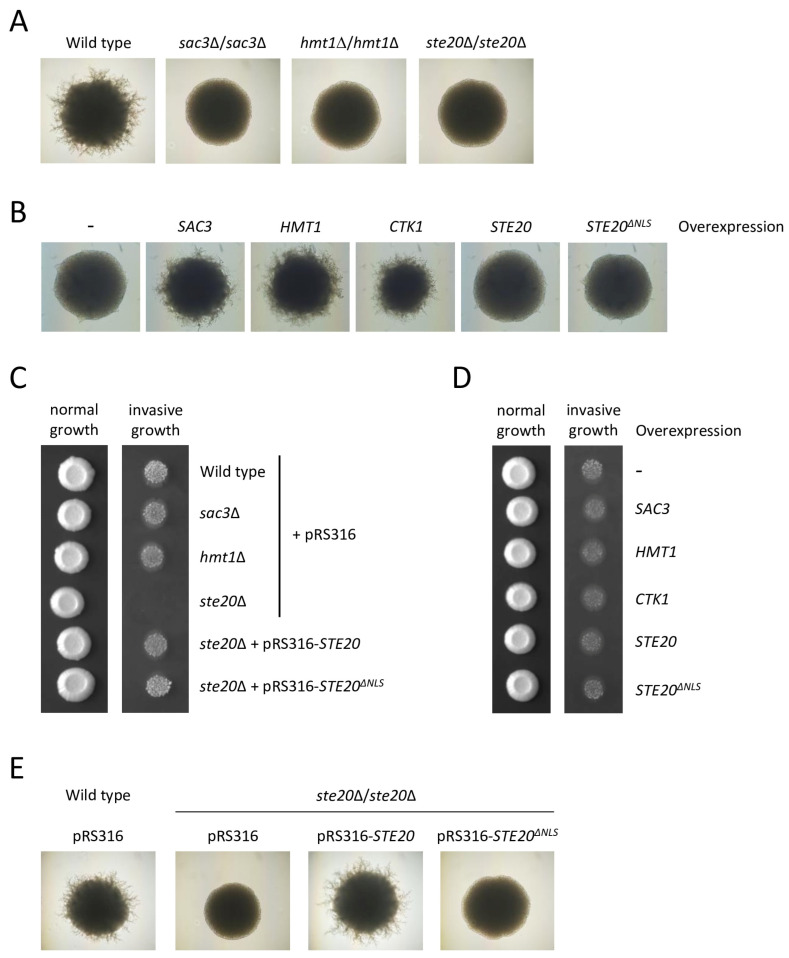
Nuclear Ste20, Sac3, Hmt1 and Ctk1 play a role in diploid pseudohyphal growth but not in haploid invasive growth. (**A**) *SAC3* and *HMT1* are essential for diploid pseudohyphal growth. The indicated strains were grown on synthetic low ammonium dextrose (SLAD) medium for 7 days. A smooth colony surface indicates the absence of pseudohyphal growth. (**B**) Overexpression of *SAC3*, *HMT1* and *CTK1* results in increased diploid pseudohyphal growth. All strains carried a multicopy pRS426 plasmid with the indicated gene. Cells were grown on low-nitrogen SLAD medium for 4 days. Images were taken at this early stage shortly after pseudohyphal growth started in the wild type to illustrate the stronger filamentous growth in strains overexpressing *SAC3*, *HMT1* and *CTK1*. (**C**) Deletion of *SAC3* and *HMT1* does not affect haploid invasive growth. In total, 10^4^ cells of the indicated strains were grown on selective medium. An image was taken after 5 days (normal growth). The plate was then washed in a stream of water and photographed again to reveal invasive growth. (**D**) Overexpression of genes encoding nuclear proteins has no effect on haploid invasive growth. Wild type cells carrying a multicopy pRS426 plasmid with the indicated genes were grown on selective medium. (**E**) Nuclear localization of Ste20 is required for diploid pseudohyphal growth. Cells of the indicated strains were grown on low-nitrogen SLAD medium for 7 days.

**Figure 8 ijms-24-15916-f008:**
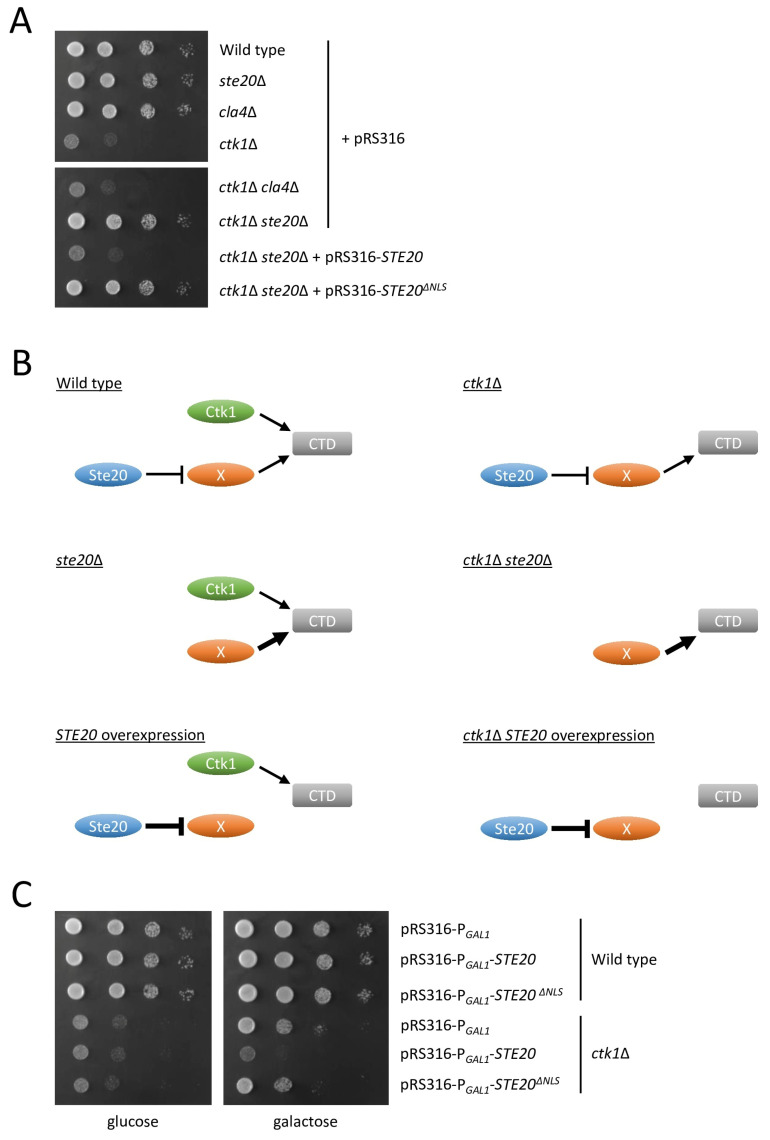
Genetic interactions between *STE20* and *CTK1*. (**A**) *STE20* deletion rescues the growth defect of cells lacking *CTK1*. Serial dilutions (1:10) of the indicated strains were grown on selective medium. (**B**) Model for genetic interactions between *STE20* and *CTK1*. Ctk1 is a major regulator of RNA II polymerase through phosphorylation of the C-terminal domain (CTD) of the largest subunit. We propose that the CTD is also modified by a hypothetical protein X. This unknown protein is negatively regulated by Ste20. Due to the critical role that Ctk1 plays, cells lacking this kinase display a major growth defect, assuming that the hypothetical protein X is less important. Activity of this protein increases in the absence of the inhibitory effect of Ste20. This does not affect the growth rate in a *ste20*Δ strain. In contrast, when both *STE20* and *CTK1* are absent, the higher activity of the hypothetical protein can compensate for the growth defect caused by the loss of *CTK1*. *STE20* overexpression results in reduced activity of the hypothetical protein. Again, this has no effect in the presence of Ctk1 but cells lacking *CTK1* exhibit an even more severe growth defect. Line weights indicate activity levels of proteins. For simplification, Ctk2 and Ctk3, the other subunits of the CTD kinase I complex, and the RNA polymerase II subunits are not shown. The model shown here is for RNA polymerase II. As described in the discussion, this could also be RNA polymerase I. (**C**) Overexpression of nuclear *STE20* exacerbates the growth defect of *ctk1*∆ cells. Serial dilutions (1:10) of the indicated strains were grown on selective medium containing glucose or galactose. The cells carried either an empty plasmid or plasmids in which *STE20* or *STE20*^Δ*NLS*^ were placed under control of the inducible *GAL1* promoter.

**Figure 9 ijms-24-15916-f009:**
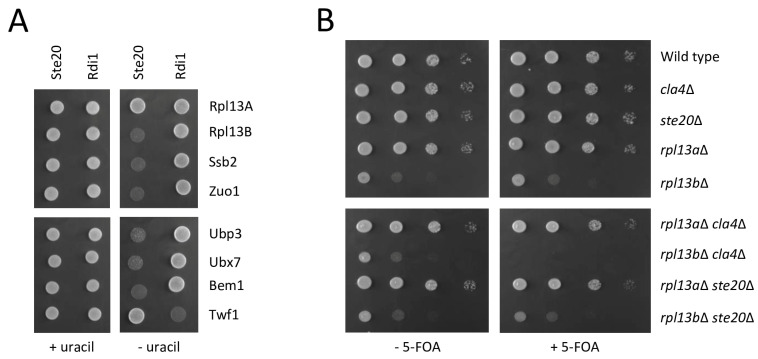
Ste20 binds to ribosome-associated proteins and factors involved in ubiquitin-dependent protein degradation. (**A**) Split-ubiquitin interactions between Ste20 and ribosome-associated proteins and proteins with ubiquitin-related functions. In this experiment, 10^4^ cells of the indicated plasmid combinations were spotted onto control plates supplemented with uracil, and onto plates lacking uracil to monitor protein–protein interactions. Bem1 and Twf1 were included as controls for Ste20 and Rdi1. (**B**) Genetic interaction between *CLA4* and *RPL13B*. Serial dilutions (1:10) of the indicated strains were spotted on selective medium with or without 5-FOA. Strains lacking *CLA4* carried a *CLA4* copy on a pRS316 plasmid. All other strains carried an empty pRS316 vector. 5-FOA selects for cells that have lost the *URA3*-based pRS316 plasmids.

**Table 1 ijms-24-15916-t001:** Proteins identified in the Ste20 split-ubiquitin screen. Data come from the *Saccharomyces* Genome Database (https://www.yeastgenome.org accessed on 1 September 2023). For simplification, subcompartments are not listed. For example, “nucleoplasm”, “nucleolus”, “kinetochore” and “CTDK-I complex” are all included in “nucleus”. A, actin cortical patch. C, cytoplasm. CW, cell wall. ER, endoplasmic reticulum. G, Golgi apparatus. LD, lipid droplet. M, mitochondrion. N, nucleus. PM, plasma membrane. R, ribosome. SPB, spindle pole body. SV, secretory vesicle. U, unknown. V, vacuole.

Protein	Molecular Function and/or Biological Process	Localization
	** Glycolysis **	
Cdc19	Pyruvate kinase	C, PM
Gpm1	Phosphoglycerate mutase	C, M
Pgk1	Phosphoglycerate kinase	C, M, PM
Tdh3	Glyceraldehyde 3-phosphate dehydrogenase	C, CW, LD, M, N, PM
	** Lipid metabolism and homeostasis **	
Ale1	Lysophospholipid acyltransferase involved in glyerophospholipid biosynthesis	ER
Cbr1	Cytochrome *b*_5_ reductase involved in ergosterol biosynthesis	N, M
Erg4	C-24(28) sterol reductase involved in ergosterol biosynthesis	ER
Hes1	Sterol transfer protein involved in exocytosis, endocytosis and maintenance of cell polarity	PM, V
Ncp1	NADPH-cytochrome P450 reductase involved in ergosterol biosynthesis	C, ER, M
Scs7	Fatty acid α-hydroxylase involved in inositol phosphoceramide metabolism	ER
Sut1	Zn(II)_2_Cys_6_ family transcription factor that regulates sterol uptake genes	C, N
	** Other metabolic pathways **	
Dfr1	Dihydrofolate reductase involved in tetrahydrofolate biosynthesis	C, M
Ima5	Oligosaccharide α-1,6-glucosidase involved in disaccharide catabolism	U
	** Nuclear functions **	
Ctk1	Catalytic subunit of C-terminal domain kinase I which phosphorylates RNA polymerase II to affect transcription and pre-mRNA 3’ end processing	N
Hat2	Histone H4 acetyltransferase involved in chromatin assembly, disassembly and silencing	C, N
Hmt1	Protein-arginine methyltransferase involved in transcription regulation and mRNA export	N
Mcm22	Kinetochore protein involved in chromosome segregation	N
Pta1	Subunit of mRNA cleavage and polyadenylation specificity factor	N
Rpa190	Subunit of RNA polymerase I involved in ribosomal RNA transcription	N
Sac3	Involved in mRNA export from the nucleus	N
Sin3	Component of histone deacetylase complexes involved in diverse processes	N
Spb1	rRNA methyltransferase involved in rRNA processing and maturation of large ribosomal subunit	N
	** Translation and protein folding **	
Efb1	Guanyl-nucleotide exchange factor subunit of the translation elongation factor 1 complex	C
Egd2	α subunit of the nascent polypeptide-associated complex	C
Ncs2	Protein involved in tRNA wobble uridine thiolation and protein urmylation	C
Rpl13B	Subunit of the large ribosomal subunit	R
Ssb2	Cytoplasmic ATPase that is a ribosome-associated molecular chaperone	C, PM, R
Tif3	Translation initiation factor eIF4B	C
Tma22	Protein of unknown function that associates with ribosomes	C, R
Zuo1	Ribosome-associated chaperone	C, M, N, R
	** Cell polarity **	
Abp1	Protein that activates actin nucleation mediated by the Arp2/Arp3 complex	A, C
Bem1	Scaffold protein involved in establishing cell polarity and morphogenesis	C, M, PM
Cdc15	Protein kinase involved in mitotic exit, cytokinesis and meiotic spindle disassembly	C, SPB
Mlc1	Myosin light chain involved in formation and contraction of actomyosin ring	PM, SV
Rvs167	Cytoskeletal protein involved in actin cytoskeleton reorganization and endocytosis	A, C
	** Ubiquitin-related functions **	
Cdc53	Subunit of Skp1/Cul1/F-box complexes required for G1/S and G2/M phase transitions	C, N
Duf1	Ubiquitin-binding protein of unknown function	C
Sgt1	Co-chaperone protein that associates with the Skp1/Cul1/F-box ubiquitin ligase complex	C, ER
Ubp3	Ubiquitin-specific protease that regulates the osmotic stress response and ER to Golgi vesicle-mediated transport	C
Ubx7	Protein of unknown function involved in ubiquitin-dependent protein degradation	N, ER
	** Hsp90 functions **	
Aha1	Co-chaperone that regulates the activity of members of the Hsp90 family	C, N
Cpr6	Peptidyl-prolyl cis-trans isomerase involved in folding of several proteins including Hsp90	C
Sse1	Adenyl-nucleotide exchange factor involved in protein folding and refolding	C
Tah1	Component of Rvb1-Rvb2-Tah1-Pih1 complex that interacts with Hsp90 to mediate assembly of large protein complexes	C, N
	** Vesicle transport **	
Arf2	GTPase that acts as ADP ribosylation factor involved in regulation of vesicle formation in ER to Golgi and Golgi to plasma membrane vesicular transport	C, G
Get3	Guanyl-nucleotide exchange factor involved in ER protein-membrane targeting and cis-Golgi to rough ER transport	C, ER
Trx2	Putative disulfide oxidoreductase involved in cellular redox homeostasis and vesicle-mediated traffic between the ER and Golgi	C
	** Vacuolar proteins **	
Vma13	Subunit of V-ATPase involved in vacuolar acidification	V
Vtc4	Polyphosphate kinase involved in vacuolar transport and vacuole fusion	ER, V
	** Transmembrane transport **	
Alr2	Inorganic cation transmembrane transporter that moves magnesium ions	PM
Atr2	Borate efflux transmembrane transporter	PM
	** Other functions **	
Efr3	Protein involved in protein-plasma membrane targeting	M, PM
Ira1	GTPase-activating protein which regulates Ras protein signal transduction	M, PM
Pps1	Protein phosphatase involved in DNA replication and MAP kinase inactivation	U
Kin82	Putative protein kinase involved in the regulation of phospholipid translocation	U
Lsb1	Protein involved in regulation of Arp2/3 complex-mediated actin nucleation	A, C, N,

**Table 2 ijms-24-15916-t002:** Published physical interactions between Ste20 and proteins identified in the split-ubiquitin screen. Data are from the *Saccharomyces* Genome Database (https://www.yeastgenome.org, accessed on 1 September 2023). PCA, protein-fragment complementation assay. *, split-ubiquitin assay.

Protein	Type of Assay
Bem1	Two-hybrid
	Affinity capture—Western
	Phosphorylation
	PCA
	Co-crystal structure
	Reconstituted complex
	PCA *
	Protein-peptide
Cbr1	Affinity capture—Western
Erg4	Affinity capture—Western
Mlc1	PCA *
Spb1	Phosphorylation
Ssb2	Co-purification
Sut1	Affinity capture—Western
Ubx7	Phosphorylation
Vma13	Affinity capture—Western

**Table 3 ijms-24-15916-t003:** Mutant phenotypes of genes identified in the Ste20 split-ubiquitin screen. Data come from the *Saccharomyces* Genome Database (https://www.yeastgenome.org, accessed on 1 September 2023). With the exception of *BEM1*, all phenotypes are for null deletion or overexpression strains. For *BEM1*, several unspecified alleles (*bem1*-*m1*, *bem1*-*m2* and *bem1*-*s1*) have also been used.

Gene	Filamentation	Pheromone Response	Hyperosmotic Stress Response
*ABP1*	Null	-	-
	Overexpression		
*ALR2*	Overexpression	-	-
*ARF2*	Overexpression	-	-
*BEM1*	Null	*bem1-s1*	Null
		*bem1-m1* and *bem1-m2*	
		Null	
*CTK1*	-	-	Null
			Null
*DFR1*	Overexpression	-	-
*EFB1*	Null	-	-
*EFR3*	Overexpression	-	-
*ERG4*	Null	Null	Null
*GET3*	-	Null	-
		Overexpression	
*HES1*	Overexpression	-	-
*IRA1*	Null	-	-
*KIN82*	-	Overexpression	-
*NCP1*	Null	-	-
	Null		
	Null		
*NCS2*	Overexpression	-	-
*PGK1*	Null	-	-
*PPS1*	Overexpression	-	-
*RPA190*	Null	-	-
*RPL13B*	-	Null	Null
*RVS167*	-	-	Null
			Null
*SAC3*	-	-	Null
*SCS7*	-	-	Null
*SIN3*	-	Null	-
*SSE1*	-	-	Null
*SUT1*	Overexpression	Null	Null
*TIF3*	-	-	Null
*UBP3*	Overexpression	-	-
*UBX7*	Overexpression	-	-
*VMA13*	Null	-	-
*ZUO1*	-	-	Null
			Null

**Table 4 ijms-24-15916-t004:** Negative genetic interactions between *CLA4* and genes identified in the *STE20* split-ubiquitin screen. Data are from the *Saccharomyces* Genome Database (https://www.yeastgenome.org, accessed on 1 September 2023).

Gene	Type of Interaction
*BEM1*	Synthetic lethality
	Negative genetic
*CDC15*	Negative genetic
*CDC53*	Negative genetic
*CTK1*	Negative genetic
*ERG4*	Synthetic lethality
*NCP1*	Synthetic lethality
*NCS2*	Synthetic lethality
*RVS167*	Synthetic growth defect
	Synthetic lethality
	Negative genetic
*SAC3*	Negative genetic
*SIN3*	Negative genetic
	Synthetic lethality
*SSE1*	Negative genetic
*TDH3*	Negative genetic

**Table 5 ijms-24-15916-t005:** Biological functions of proteins interacting with Ste20. Interactors identified by others are from the *Saccharomyces* Genome Database (https://www.yeastgenome.org, accessed on 1 September 2023).

Biological Process	Ste20 Interactors Identified Here	Ste20 Interactors Identified by Others
Lipid metabolism and homeostasis	Ale1, Cbr1, Erg4, Hes1, Ncp1, Scs7, Sut1	Are2, Cho2, Cst26, Pah1, Sct1
Other metabolic pathways	Cdc19, Dfr1, Gpm1, Ima5, Pgk1, Tdh3	Pcm1, Trp2, Yig1
Ubiquitin-related functions	Cdc53, Duf1, Sgt1, Ubp3, Ubx7	Aly2, Rpt5, Ubp7, Ubp9, Ubx5, Ubx7
Vesicle transport and exocytosis	Arf2, Get3, Trx2	Apl5, Boi1, Boi2, Cog4, Ent2, Erv46, Exo84, Glo3, Msb4, Sec2, Sro7
Cell polarity	Abp1, Bem1, Cdc15, Mlc1, Rvs167	Aim3, Bem1, Bem4, Bni1, Bud6, Bud8, Cbk1, Cdc3, Cdc24, Cdc42, Rga2, Mlc1, Myo3
MAPK signaling	-	Nbp2, Ptc1, Ptp2, Sho1, Slt2, Ste11
tRNA synthesis, modification and export	Pta1	Ats1, Dus1, Gus1, Krs1, Ses1, Sol2
Ribosome biogenesis	Rpa190, Spb1	Bms1, Drs1, Erb1, Nsi1, Nsr1, Rcm1, Rnh70, Rrn6, Sas10, Spb1, Utp5, Utp7
Translation and protein folding	Efb1, Egd2, Ncs2, Rpl13B, Ssb2, Sse1, Tif3, Tma22, Zuo1	Asc1, Cdc33, Ebs1, Rpl25, Rpl26A, Rps1A, Rps2, Rps3, Rps6A, Rps7A, Rps8A, Rps9A, Rps10A, Rps12, Rps13, Rps14A, Rps15, Rps19A, Rps20, Rps27A, Sbp1, Ssb1, Ssb2, Ssd1
Chromatin organization	Hat2, Sin3	Cse4, Hhf1, Htb1, Htb2, Isw1, Rsc6, Rsc8, Spt16
Transcription	Ctk1	Azf1, Bdf2, Brf1, Ccr4, Cin5, Ckb1, Crz1, Rpb5, Sgv1
Other nuclear functions	Hmt1, Mcm22, Pps1	Mpc54, Mps3, Mrc1, Rad1, Rad53, Spo21
mRNA binding	Sac3	Bfr1, Crm1, Dcp2, Dhh1, Gsp1, Hek2, Nab2, Nup53, Prp21, Puf4, Rim4, She3, Tho1, Tom1
Cell cycle	-	Cdc28, Clb1, Clb2, Clb5, Cln1, Cln2, Hsl7
Other functions	Aha1, Alr2, Atr2, Cpr6, Efr3, Ira1, Kin82, Lsb1, Tah1, Vma13, Vtc4	Atg9, Avt4, Bmh1, Bmh2, Cnm1, Ecm3, Ema35, Frt2, Hsp30, Ist2, Mdg1, Mge1, Mrpl36, Mrx9, Nvj2, Pam1, Sdd3, Stb2, Ste4, Ybr071w, Yhr131c, Yml037c, Yml083c, Yml096w

**Table 6 ijms-24-15916-t006:** Yeast strains used in this study.

Strain	Genotype	Source
AMY2	PC344 *sac3*∆::*hphNT1*/*sac3*∆::*KanMX6*	This study
CTY64	YPH499 *rdi1*∆::*KanMX6*	[131]
MLY208	YPH499 *STE20*-*3HA*-*His3MX6*	This study
MLY215	PPY966 *hmt1*∆::*hphNT1*	This study
MLY219	PC344 *hmt1*∆::*hphNT1*/*hmt1*∆::*KanMX6*	This study
MLY224	YPH499 *sac3*∆::*His3MX6*	This study
MLY225	YPH499 cla4∆::*klTRP1 sac3*∆::*His3MX6*	This study
MLY227	YPH499 *sac3*∆::*His3MX6* ste20∆::*klTRP1*	This study
MLY241	YPH499 *ctk1*∆::*KanMX6* ste20∆::*klTRP1*	This study
MLY255	PPY966 *sac3*∆::*His3MX6*	This study
PC344	*MAT*a/*MATα ura3-52*/*ura3*-*52*	[131]
PPY966	*MAT*a *his3*::*hisG leu2*::*hisG trp1*::*hisG ura3*-*52*	[32]
SHY93	YPH499 *ctk1*∆::*His3MX6*	This study
THY310	YPH499 ste20∆::*klTRP1*	[32]
THY608	YPH499 cla4∆::*klTRP1*	[17]
THY697	PPY966 *ste20*∆::*hphNT1*	[131]
THY706	PC344 *ste20*∆::*hphNT1*/*ste20*∆::*KanMX6*	[131]
THY878	YPH499 *STE20*-*3HA*-*His3MX6* *PGK1*-*9Myc*-*KanMX6*	This study
THY879	YPH499 *rpl13a*∆::*His3MX6*	This study
THY880	YPH499 *rpl13a*∆::*His3MX6* ste20∆::*klTRP1*	This study
THY883	YPH499 cla4∆::*klTRP1 rpl13a*∆::*His3MX6*	This study
THY885	YPH499 *sem1*∆::*His3MX6*	This study
THY886	YPH499 *sem1*∆::*His3MX6* ste20∆::*klTRP1*	This study
THY887	YPH499 ste20∆::*klTRP1 sus1*∆::*His3MX6*	This study
THY888	YPH499 *sus1*∆::*His3MX6*	This study
THY889	YPH499 *thp1*∆::*KanMX6*	This study
THY890	YPH499 ste20∆::*klTRP1 thp1*∆::*KanMX6*	This study
THY891	YPH499 cla4∆::*klTRP1 sem1*∆::*His3MX6*	This study
THY892	YPH499 cla4∆::*klTRP1 sus1*∆::*His3MX6*	This study
THY894	YPH499 cla4∆::*klTRP1 thp1*∆::*KanMX6*	This study
THY895	YPH499 *rpl13b*∆::*His3MX6*	This study
THY896	YPH499 ste20∆::*klTRP1 rpl13b*∆::*His3MX6*	This study
THY897	YPH499 cla4∆::*klTRP1 rpl13b*∆::*His3MX6*	This study
YPH499	*MAT*a *ura3*-*52* *lys2*-*801 ade2*-*101 trp1*∆*63 his3*∆*200 leu2*∆*1*	[132]

**Table 7 ijms-24-15916-t007:** Plasmids used in this study. *STE20*^∆*NLS*^ constructs lack base pairs 814–862 of the *STE20* open reading frame encoding the NLS (residues 272–288) [6].

Plasmid	Genotype	Source
pADNX	2 μm, *LEU2*, P*_ADH1_-NUbiquitin-HA*	[133]
pAK4	pADNX carrying *GPM1*	This study
pAK7	pADNX carrying *PGK1*	This study
pCT2	pRS313 carrying P*_MET25_*-*RDI1*-*CUbiquitin*-*RURA3*	[17]
pIJ1	pADNX carrying *GND2*	This study
pIJ2	pADNX carrying *FBP1*	This study
pIJ3	pADNX carrying *GLK1*	This study
pIJ4	pADNX carrying *SOL4*	This study
pKA86	pRS316 carrying P*_GAL1_*-*STE20*-*3HA*	[17]
pML70	pRS313 carrying P*_MET25_*-*CLA4*-*CUbiquitin*-*RURA3*	[33]
pML110	pRS426 carrying *HMT1*	This study
pML112	pADNX carrying *SAC3*	This study
pML118	pRS313 carrying P*_MET25_*-*STE20*^∆*NLS*^-*CUbiquitin*-*RURA3*	This study
pRS313	*CEN, HIS3*	[132]
pRS316	*CEN, URA3*	[132]
pRS426	2 μm, *URA3*	[134]
pTH102	pRS316 carrying *CLA4*	[32]
pTH197	pRS313 carrying P*_MET25_*-*STE20*-*CUbiquitin>*-*RURA3*	[32]
pTH256	pRS316 *carrying STE20*	[32]
pTH344	pADNX *carrying BEM1*	[32]
pTH351	pADNX *carrying CDC19*	This study
pTH369	pADNX *carrying CTK1*	This study
pTH371	pADNX *carrying HMT1*	This study
pTH418	pADNX *carrying TDH3*	This study
pTH424	pADNX *carrying SOL3*	This study
pTH425	pADNX *carrying GND1*	This study
pTH426	pADNX *carrying ZWF1*	This study
pTH427	pADNX *carrying HXK2*	This study
pTH428	pADNX *carrying TKL1*	This study
pTH430	pADNX *carrying PGI1*	This study
pTH431	pADNX *carrying PFK1*	This study
pTH432	pADNX *carrying FBA1*	This study
pTH434	pADNX *carrying ENO1*	This study
pTH437	pADNX *carrying TPI1*	This study
pTH438	pADNX *carrying TDH1*	This study
pTH439	pADNX *carrying RKI1*	This study
pTH440	pADNX *carrying RPE1*	This study
pTH443	pADNX *carrying PCK1*	This study
pTH444	pADNX *carrying PFK2*	This study
pTH448	pADNX *carrying ENO2*	This study
pTH449	pADNX *carrying TDH2*	This study
pTH458	pADNX *carrying LEU2*	This study
pTH459	pADNX carrying *TAL1*	This study
pTH465	pADNX carrying *UBP3*	This study
pTH466	pADNX carrying *TKL2*	This study
pTH467	pADNX carrying *TWF1*	This study
pTH469	pRS426 carrying *CTK1*	This study
pTH470	pADNX carrying *HXK1*	This study
pTH471	pADNX carrying *PYK2*	This study
pTH472	pADNX carrying *CDC42*	This study
pTH473	pADNX carrying *PYC1*	This study
pTH474	pADNX carrying *PYC2*	This study
pTH475	pADNX carrying *THP1*	This study
pTH476	pADNX carrying *SEM1*	This study
pTH477	pADNX carrying *SUS1*	This study
pTH478	pADNX carrying *CDC31*	This study
pTH479	pADNX carrying *RPL13A*	This study
pTH480	pADNX carrying *RPL13B*	This study
pTH482	pADNX carrying *ZUO1*	This study
pTH483	pADNX carrying *UBX7*	This study
pTH485	pADNX carrying *SAC3*^*901*−*1301*^	This study
pTH487	pADNX carrying *SSB2*	This study
pTH488	pRS316 carrying *STE20*^∆*NLS*^	This study
pTH489	pRS426 carrying *SAC3*	This study
pTH490	pRS426 carrying *STE20*	This study
pTH491	pRS426 carrying *STE20*^∆*NLS*^	This study
pTH492	pRS316 carrying P*_GAL1_*-*STE20*^∆*NLS*^-*3HA*	This study
pTH494	pADNX carrying *SAC3*^*1001*−*1301*^	This study
pTH495	pADNX carrying *SAC3*^*1101*−*1301*^	This study
pTH496	pADNX carrying *SAC3*^*1201*−*1301*^	This study
pTH501	pADNX carrying *SAC3*^*1251*−*1301*^	This study

## Data Availability

All data underlying the results are available as part of the article.

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
