# Peer review of "A Protein–Protein Interaction Analysis Suggests a Wide Range of New Functions for the p21-Activated Kinase (PAK) Ste20"

_ijms, 2023, doi:10.3390/ijms242115916_

Round 1
Reviewer 1 Report
Comments and Suggestions for Authors
Comments to Authors
MSSREF IJMS-2649271
In the present manuscript entitled “A protein-protein interaction analysis suggests a wide range of 2 new functions for the p21-activated kinase (PAK) Ste20” Ifeoluwapo Matthew Joshua et al., described the protein-protein interaction of Ste20 with 56 different proteins which has not been identified earlier and belong to vesicle transport and translation. Ste20 is proposed to play critical role in stimulation of mitogen-activated protein kinase (MAPK) pathways which eventually leads to transcription factor activation.
This is a well written and organized but bit complex MSS to follow.
Results and discussion section are well presented but can be narrow down however introduction is lacking some information. Last two paragraphs of introduction are very strong statement and need bit more elaboration and that issue has not been discussed in discussion section, if I am correct. The condition stated in this section are very distinct and complex pathologies and Ste20 interaction with several interacting proteins identified in this study are associated with any pathological condition directly is not clear. It is possible that this reviewer is missing such information in text.
It is difficult to understand the purpose of section 2.1 in result section, this need some simple explanation that is easily understandable for general reader.
What are the critical determinant players for this interaction in lack of binding as stated on page 20 and 21.
if there is no binding, what support authors have which supplement binding and support the concept of interaction.
In line 552-554 authors identified some of the techniques in support of interaction, however none of these techniques confirmed protein-protein interaction rather support the possibility of interaction. whether techniques including co-immunoprecipitations, pulldown assays, enzyme activity assays, co-localization are sufficient enough to support interaction is difficult to say.
In line 574-575 As stated functional overlapping may not be good reason for the physical interaction, several factor might play crucial role here.
As stated in line 599-601, these are well integrated events and modulated by several factors and need some better explanation.
Discussion section is very descriptive and speculative for no reason and need to be shorten and conclusive.
In section 4.3 of material and methods authors wrote “100 ml cells were grown to an A600 of 1” is not clear for this reviewer. Please make it simple and clear for general readers.
Number of references are relatively too many.
Author Response
We would like to thank the Reviewer for their constructive feedback. We made the requested changes and feel that as a consequence the quality of the manuscript has increased considerably.
“This is a well written and organized but bit complex MSS to follow.
Results and discussion section are well presented but can be narrow down however introduction is lacking some information. Last two paragraphs of introduction are very strong statement and need bit more elaboration and that issue has not been discussed in discussion section, if I am correct. The condition stated in this section are very distinct and complex pathologies and Ste20 interaction with several interacting proteins identified in this study are associated with any pathological condition directly is not clear. It is possible that this reviewer is missing such information in text.”
We agree that this is a complex manuscript and that the discussion is quite long. We have now simplified it and shortened the discussion. This is described in more detail below.
We also agree that the statement at the end of introduction was too strong and possibly suggesting that there is a link between budding yeast Ste20 and several human diseases. This has now been rephrased (“Due to the high conservation of PAKs, our results may also be relevant for PAKs in higher eukaryotes. In humans, PAKs are implicated in cancer, infectious diseases, diabetes, neurological disorders and cardiac diseases [21,22]. Because of these links, a better understanding of PAK biology becomes increasingly important.”) (lines 63-66).
“It is difficult to understand the purpose of section 2.1 in result section, this need some simple explanation that is easily understandable for general reader.”
In section “2.1 Overview of the screen” we summarize the results of the screen as highlighted in the heading of this section. The first few paragraphs describe the experimental approach for the screen (first paragraph, lines 69-85), the advantages of this technique (second paragraph, lines 86-91), mention the number of proteins that were identified in the screen and mention that some of these interactors have also been described by others (third paragraph, lines 92-103) and finally it is mentioned that a few of the proteins identified here, have already been characterized by us before (fourth paragraph, lines 114-119). We feel that this section is typical for the description of a screen and self-explanatory.
However, we agree with the Reviewer that the following sections are less clear. We have, therefore, added the following explanations for clarification.
“To find out whether these interactions are physiologically relevant, we further analyzed them also including published observations. First, the interactions identified in the screen presented here seem to be highly specific…” (lines 120-122)
“The localization of the proteins identified here also suggests that the interactions are real. Importantly, all of the Ste20 interactors can be found in compartments where they could physically interact with Ste20.” (lines 132-134)
“What are the critical determinant players for this interaction in lack of binding as stated on page 20 and 21.”
It is not quite clear to us what the determinants are. Since the discussion is already quite long we felt it would be best to abstain from further speculation.
“In line 552-554 authors identified some of the techniques in support of interaction, however none of these techniques confirmed protein-protein interaction rather support the possibility of interaction. whether techniques including co-immunoprecipitations, pulldown assays, enzyme activity assays, co-localization are sufficient enough to support interaction is difficult to say.”
This section has overall been shortened and the statement above has now been removed (“(2) Interactions between Ste20 and 5 proteins (Cbr1, Erg4, Ncp1, Sut1 and Vma13) described in this screen have previously been characterized by us [14,17,32,33].”) (lines 545-546).
“In line 574-575 As stated functional overlapping may not be good reason for the physical interaction, several factor might play crucial role here.”
The mutant phenotypes mentioned here indicate that the majority of proteins identified in the screen, like Ste20 play a role in hyperosmotic stress response, pheromone response and/or filamentous growth. We agree with the reviewer, that this simply means that Ste20 and the proteins identified by us are involved in the same biological process. This alone of course does not mean that these proteins and Ste20 physically interact. The mutant phenotypes are only one of seven reasons listed in this section (lines 544-569) which are consistent with interactions between Ste20 these proteins. However, we feel that all seven reasons taken together suggest that the interactions are physiologically relevant.
“As stated in line 599-601, these are well integrated events and modulated by several factors and need some better explanation.”
In this section, we simply state that proteins interacting with Ste20 fall into a small number of functional groups. We agree with the reviewer that these are complex processes. We, therefore, suggest that more research should be done in future, rather then giving an explanation.
“Like the Ste20 interactors identified by us, the huge majority of the proteins that physically interact with Ste20 described by others can also easily be grouped together according to their functions (Table 5). The functional categories for the Ste20 interactors described by us and by other groups show considerable overlap (Tables 1 and 5). Some of these functions include well established roles for Ste20 such as MAPK signaling and cell polarity [4,110]. However, other functions such as ribosome biogenesis and tRNA synthesis and modification are rather unexpected and these new links are certainly worth to be examined in the future.” (lines 577-584)
“Discussion section is very descriptive and speculative for no reason and need to be shorten and conclusive.”
We agree that the discussion was quite long. We have now removed the most speculative and descriptive parts. The discussion has been substantially shortened by 748 words. A conclusion can be found in lines 655-677 (“What does all this mean?...”).
“In section 4.3 of material and methods authors wrote “100 ml cells were grown to an A600 of 1” is not clear for this reviewer. Please make it simple and clear for general readers.”
This has now been rephrased and is clearer (“100 ml cells in the exponential phase were grown until they reached an optical density of 1 (measured at 600 nm).”) (lines 720-721).
“Number of references are relatively too many.”
We have now reduced the number of references substantially from 191 to 134. In particular, references from Tables 2,3 and 4 have been removed. As mentioned in the table legends, these references can easily be found in the Saccharomyces Genome Database (SGD).
Reviewer 2 Report
Comments and Suggestions for Authors
Authors of the manuscript entitled "A protein-protein interaction analysis suggests a wide range of new functions for the p21-activated kinase (PAK) Ste20" describe the identification of a number of previously unknown interacting partners of p21-activated kinase Ste20 by a split-ubiquitin system in yeast. Based on the identity of newly identified proteins they suggest the roles for Ste20 in several cellular processes, such as filamentous growth and others.
Manuscript is written in good English and carefully edited. Experiments are well designed and the results appear to be reasonably interpreted.
I did not detect any flaws that would prevent the publication of the manuscript in this form.
Round 2
Reviewer 1 Report
Comments and Suggestions for Authors
No Further comments